# A COMPUTATIONAL FRAMEWORK TO UNIFY REPRESENTATION SIMILARITY AND FUNCTION IN BIOLOGICAL AND ARTIFICIAL NEURAL NETWORKS

## ABSTRACT

Artificial neural networks (ANNs) are powerful tools for studying neural representations in the ventral visual stream of the brain, which, in turn, have inspired new designs of ANN models to improve task performance. However, a unified framework for merging these two directions has been lacking so far. In this study, we propose an integrated framework called Deep Autoencoder with Neural Response (DAE-NR), which incorporates information from the visual cortex into ANN models to achieve better image reconstruction performance and higher neural representation similarity between biological and artificial neurons. The same visual stimuli (*i.e.*, natural images) are input to both the mice brain and DAE-NR. The encoder of DAE-NR jointly learns the dependencies from neural spike encoding and image reconstruction. For the neural spike encoding task, the features derived from a specific hidden layer of the encoder are transformed by a mapping function to predict the ground-truth neural response under the constraint of image reconstruction. Simultaneously, for the image reconstruction task, the latent representation obtained by the encoder is assigned to a decoder to restore the original image under the guidance of neural information. In DAE-NR, the learning process of encoder, mapping function and decoder are all implicitly constrained by these two tasks. Our experiments demonstrate that *if and only if* with the joint learning, DAE-NRs can improve the performance of visual image reconstruction and increase the representation similarity between biological neurons and artificial neurons. The DAE-NR offers a new perspective on the integration of computer vision and neuroscience.

## 1 INTRODUCTION

Computer vision has achieved almost comparable performance to the human visual system on some tasks, mainly thanks to recent advances in deep learning. Image reconstruction is one of the essential tasks in computer vision Hinton & Salakhutdinov (2006); Kingma & Welling (2014); Ravishankar et al. (2020). As a solution, the auto-encoder (AE) framework embeds the high-dimensional input to a low-dimensional latent space by the encoder and then reconstructs the image by the decoder Hinton & Salakhutdinov (2006); Goodfellow et al. (2016). Despite the popularity and the practical successes of AE models, the setting of prior would largely influence the image reconstruction performance on DAEs Tomczak & Welling (2018). Moreover, there usually needs to be more biological interpretability in the model architecture.

Inspired by neuroscience, computer vision researchers have been interested in how to use information from biological neurons to achieve brain-like performance (such as robustness and ability to learn from small samples). The biology-inspired AE models may help improve performance in image reconstruction tasks and bring biological interpretability Federer et al. (2020); Schrimpf et al. (2018); Safarani et al. (2021). To this end, the key question is how to integrate biological information into AEs.

On the other hand, computational neuroscience is interested in building models that map stimuli to neural responses. Traditional models have difficulty expressing nonlinear characteristics between stimulus and neural response. Deep learning empowers computational neuroscience models and reveals the relationship between stimuli and neural spikes Klindt et al. (2017) Although biological and

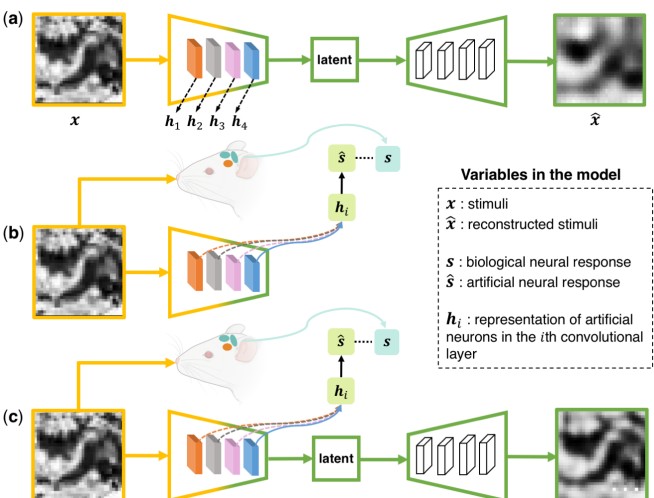

Figure 1: Scheme of the models: (a) the standard deep autoencoder (DAE) for image reconstruction, (b) the convolutional neural network with factorized readout (CNN-FR) for predicting neural responses, (c) the DAE with neuron response (DAE-NR) for both image reconstruction and neural response prediction. $s$ denotes the biological neural response; $\hat{s}$ denotes the prediction of biological neural response; $\mathbf{h}_i$ ($i \in \{1, 2, 3, 4\}$) is the feature map of the $i$th convolutional layer.

artificial neural networks may have fundamental differences in computation and learning Macpherson et al. (2021), both are realized by interconnected neurons: the former by biological neurons; the latter by artificial neurons. Some previous work has focused on investigating the similarities between the information representation of biological neurons and artificial neurons using end-to-end ANNs, suggesting that artificial neurons in different layers of ANNs share similar representations with biological neurons in brain regions along the ventral visual pathway DiCarlo et al. (2012); Yamins & DiCarlo (2016); Walker et al. (2019); Bashivan et al. (2019). How to build artificial neural networks most similar to biological neural representations remain an open question.

Some studies have leveraged real neural responses to speed up the training process and improve network performance in object detection tasks Federer et al. (2020); Schrimpf et al. (2018); Safarani et al. (2021). However, no model has yet utilized neural responses as constraints to improve image reconstruction performance. More importantly, to our knowledge, there is no unified framework that can leverage both neural responses to improve model performance and image reconstruction tasks to improve representational similarity between biological and artificial neurons.

In this paper, we aim to tackle these two questions in one piece. Specifically, we propose a united biologically inspired framework, which jointly learns i) to project features of visual input in a specific layer of the encoder to biological neural responses by a mapping function and ii) to reconstruct the visual input via the decoder. As a result, its encoder has a higher representational similarity to the real neural responses, and its decoder achieves better image reconstruction performance. Our contributions can be summarized as follows.

- We present a biologically inspired framework called Deep Autoencoder with Neural Response (DAE-NR). The framework can simultaneously learn to predict neural responses and to reconstruct the visual stimuli (**Sec. 3**).

- The deep autoencoders embedded in DAE-NR can improve the image reconstruction quality with the help of a Poisson loss on the predicted neural activity, compared to the baseline auto-encoder models (**Sec. 4.2 & 4.4**).

- The computational neuroscience model (CNM) via DAE-NR offers higher resemblance between artificial neurons and biological neurons compared to the competing end-to-end CNM models (**Sec. 4.3 & 4.4**).

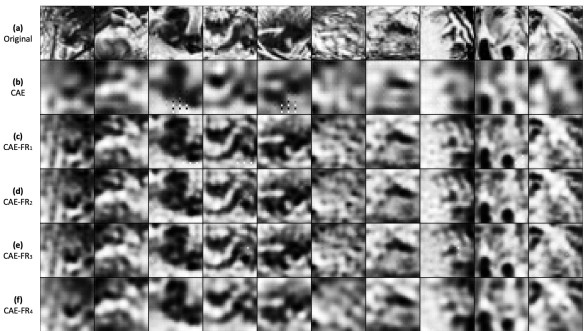

Figure 2: Reconstructed images of neural activity in mouse Region 3. From top to bottom, each row displays 10 examples of the original stimuli images (a), the reconstructed images by CAE (b), and by CAE-FR$_i$ (c-f) for $i \in \{1, 2, 3, 4\}$, respectively.

## 2 RELATED WORK

**Image reconstruction via auto-encoders:** A big breakthrough in image reconstruction is in Hinton & Salakhutdinov (2006), which equips an autoencoder with a stack of restricted Boltzmann machines. The denoising autoencoder Vincent et al. (2008) and convolutional autoencoder (CAE) Masci et al. (2011) further improve the stability of autoencoders by adding noise and taking advantage of convolutional layers in feature extraction, respectively. Variational Autoencoder (VAE) enhances model robustness of generating insightful representations through a latent distribution learning mechanism Kingma & Welling (2014). Vector quantisation VAE (VQ-VAE) employs the vector quantisation to obtain a discrete latent representation that can improve the quality of image reconstruction and generation van den Oord et al. (2017). Recent advances follow a similar trend of latent space exploration and generalize variants of DAE to many domains Larsen et al. (2016); Khattar et al. (2019); Park et al. (2020); Wang et al. (2021); Cai et al. (2021); Ran et al. (2021). However, DAE and its variants in image reconstruction suffer from the same problem; the parameters have a high degree of freedom. In other words, parameters can only be learned through error backpropagation guided by gradients. Therefore, some meaningful constraints on the parameter space will be beneficial for learning.

**Neural similarity in computational neuroscience:** Many models in computational neuroscience have been proposed to build a relationship between stimuli and the corresponding neural spike responses, which are known as neural spike encoding. The goal of these models is to increase the similarity between predicted and true neural responses. Historically, much effort has been devoted to finding tuning curves for specific features of visual stimuli, such as the orientation of bars, to predict neural responses (Carandini et al., 2005; Dräger, 1975; Hubel & Wiesel, 1962; 1968; Niell et al., 2008). This is feasible for some neurons in the primary visual cortex, but not for all neurons. Nonlinear methods provide a more general approach to predicting neural responses, including energy models Hubel & Wiesel (1962), the linear-nonlinear (LN) model and the LN-LN model Meyer et al. (2017). Traditional machine learning methods, such as generalized linear models (GLMs) (Willmore et al., 2008), the multi-layer perceptron (MLP) and the support vector regression (SVR) Das et al. (2019), have been widely applied in computational neuroscience to predict the neural responses to stimuli. More recently, hierarchical structures have been found in both ventral visual pathway (DiCarlo et al., 2012; Vintch et al., 2015; Rowekamp & Sharpee, 2017) and in deep convolutional neural networks (Fukushima et al., 1983; LeCun et al., 1989; Krizhevsky et al., 2012). Underlying the similar hierarchy, the brain and CNN are believed to share similar neural representations for extracting features from stimuli layer by layer, from simple to complex. To test this hypothesis, Yamins el al Yamins & DiCarlo (2016) proposed to use hierarchical CNNs as computational neuroscience models to investigate neural similarity. They report neural representation similarities between biological neurons along the ventral visual stream and artificial neurons in different convolutional layers Yamins & DiCarlo (2016). Furthermore, CNN with a fully-connected readout layer (CNN-FC) can directly map the convolutional features to neural responses McIntosh et al. (2016). However, such a fully-connected readout layer typically contains a large number of parameters. To reduce the number of parameters in the readout layer, CNN with a factorized readout

layer (CNN-FR) and CNN with a fixed mask (CNN-FM) are proposed which factorize convolutional features into a spatial mask Klindt et al. (2017). So far, CNN models with readout layers have become an important tool for studying neural similarity between the brain and CNN.

## 3 METHOD

Here we first introduce the notation of data spaces and variables. We then present the framework of deep autoencoders with neural response (DAE-NR) in Sec. 3.1. Finally, we describe realizations of the DAE-NR framework using CAE and CNN-FR in Sec. 3.2.

The data space of visual stimuli, the neural responses, and the features of stimuli in the $i$-th convolutional layer are denoted by $\mathcal{X}$, $\mathcal{S}$, and $\mathcal{H}_i$. The visual stimuli (*i.e.*, natural images) and the corresponding neural responses (*i.e.*, neural spikes in the V1 region) are represented as $\boldsymbol{x} \in \mathbb{R}^{N \times P \times P \times C}$ and $\boldsymbol{s} \in \mathbb{R}^{N \times M}$, respectively. The features of stimuli in the $i$-th convolutional layer are denoted as $\boldsymbol{h}_i \in \mathbb{R}^{N \times K \times K \times F}$, with the sample size ($N$), the image resolution ($P$), the image channel ($C$), the kernel size of a feature of stimuli ($K$), the number of kernels ($F$), and the number of V1 neurons ($M$), respectively.

### 3.1 THE DAE-NR FRAMEWORK

The DAE-NR combines the function of DAE for image reconstruction and the role of CNM for predicting neural responses. The DAE-NR framework consists of three parts, including an encoder $f_1 \colon \mathcal{X} \to \mathcal{H}_i$, a decoder $f_2 \colon \mathcal{H}_i \to \hat{\mathcal{X}}$, and a mapping function $f_3 : \mathcal{H}_i \to \hat{\mathcal{S}}$.

**DAE:** The standard DAEs and their variants (*e.g.*, CAE Masci et al. (2011), VAE (Kingma & Welling, 2014) and VQ-VAE (van den Oord et al., 2017)) consist of an encoder, the latent space, and a decoder, which have become mainstream models for image reconstruction. In our work, instead of separating the encoder and decoder by the latent layer, we split the encoder and decoder at the $i$-th layer of DAE. In this way, the architecture of our DAE turns out to be: (i) an encoder $f_1 : \mathcal{X} \to \mathcal{H}_i$, to embed the input to neural representation in $i$-th layer; (ii) a decoder $f_2 : \mathcal{H}_i \to \hat{\mathcal{X}}$, to reconstruct the input based on neural representations in $i$-th layer. We formally describe them as follows:

$$\text{the encoder: } \boldsymbol{h}_i = f_1(\phi_1, \boldsymbol{x}),$$
$$\text{the decoder: } \hat{\boldsymbol{x}} = f_2(\phi_2, \boldsymbol{h}_i), \tag{1}$$

where the $\hat{\boldsymbol{x}}$ is reconstructed from the original image $\boldsymbol{x}$. Both the encoder and decoder are realized by neural networks with parameters $\phi_1$ and $\phi_2$, respectively. The goal of DAE is to reconstruct the image by optimizing the the loss function $\mathcal{L}(\phi_1, \phi_2)$.

**CNM:** The computational neuroscience models (*e.g.*, CNN-FR (Klindt et al., 2017; Bashivan et al., 2019), CNN-FC (McIntosh et al., 2016), and CNN-FM (Klindt et al., 2017)) usually consist of the encoder and the readout layer. The encoder can be used to extract the image's features. The readout layer is used to map the feature space $\mathcal{H}_i$ to the neural responses in the space $\mathcal{S}$. The mapping function $f_3 : \mathcal{H}_i \to \hat{\mathcal{S}}$ is

$$\hat{\boldsymbol{s}} = f_3(\boldsymbol{h}_i, \theta). \tag{2}$$

We employ the $\mathcal{L}(\theta)$ loss to optimize the representation similarity between artificial neurons (*i.e.*, $\hat{\boldsymbol{s}}$) and biological neurons neurons (*i.e.*, $\boldsymbol{s}$) in the mapping function.

**DAE-NR:** The loss function of DAE-NR explicitly considers both the image reconstruction task in computer vision and the neural representation similarity task in computational neuroscience, as defined in Eq. equation 3.

$$\mathcal{L}(\phi_1, \phi_2, \theta) = \alpha * \mathcal{L}(\phi_1, \phi_2) + \beta * \mathcal{L}(\theta), \tag{3}$$

where $\alpha$ and $\beta$ are the hyperparameters to trade off the image reconstruction task and the neural representation similarity task.

### 3.2 REALIZATIONS OF DAE-NR

Here, we first realize DAE-NR by a toy model (*i.e.*, CAE-FR) combining a convolutional autoencoder (CAE) and a CNN with factorized readout (CNN-FR). It is important to note that the DAE-NR framework is compatible with different instantiations. We have full implementation in Table 1.

Table 1: The variants of DAE-NR. The implementation details of CAE-FR are presented in Sec 3.2. The implementation of other variants is the same as CAE-FR. CAE, VAE, and VQ-VAE are the short name for convolutional autoencoder, variational autoencoder, and vector quantisation VAE, respectively. $i \in \{1, 2, 3, 4\}$ representing the DAE-NR$_i$ extracted features from the different convolutional layers $\mathbf{h}_i$.

| DAEs
CNMs | CAE | VAE | VQ-VAE |
|---|---|---|---|
| CNN with factorized readout (CNN-FR) | **CAE-FR**$_i$ | VAE-FR$_i$ | VQ-VAE-FR$_i$ |
| CNN with fully-connected readout (CNN-FC) | CAE-FC$_i$ | VAE-FC$_i$ | VQ-VAE-FC$_i$ |
| CNN with fixed mask (CNN-FM) | CAE-FM$_i$ | VAE-FM$_i$ | VQ-VAE-FM$_i$ |

**CAE:** Both the encoder and the decoder are realized by convolutional neural networks with parameters $\phi_1$ and $\phi_2$, respectively. The loss function of CAE is formulated as a $L_2$ norm:

$$\mathcal{L}(\phi_1, \phi_2) = \|\boldsymbol{x} - \hat{\boldsymbol{x}}\|_2^2 = \|\boldsymbol{x} - f_2(\phi_2, f_1(\phi_1, \boldsymbol{x}))\|_2^2, \qquad (4)$$

where the $\hat{\boldsymbol{x}}$ is reconstructed from the original image $\boldsymbol{x}$.

**CNN-FR:** The CNN-FR consists of two parts, the convolutional layers as the encoder and the factorized readout layer (Klindt et al., 2017; Bashivan et al., 2019; Cadena et al., 2019; Zhuang et al., 2021). The convolutional layers convolve the image with a number of kernels followed by batch normalization, resulting in multiple feature maps. The readout layer pools the output of the convolutional layer (*i.e.*, $\boldsymbol{h_i}$) by applying a sparse mask on each neuron. Applying a sparse mask, the readout layer pools the output of the convolutional layer (*i.e.*, $\boldsymbol{h_i}$) on each neuron. Let us denote that $\boldsymbol{h}_i$ lies in the feature space $\mathcal{H}_i$ and the neural responses in the space $\mathcal{S}$. The mapping function is

$$\hat{\boldsymbol{s}} = f_3(\boldsymbol{h}_i, \theta_{\mathrm{s}}, \theta_{\mathrm{d}}) = \left[ \sum (\theta_{\mathrm{s}} \cdot \boldsymbol{h}_i) \right] * \theta_{\mathrm{d}} + \boldsymbol{b}, \qquad (5)$$

where $\theta_{\mathrm{s}}$ is the spatial mask, $\theta_{\mathrm{d}}$ is the weights sum of all features $\boldsymbol{h}_i$, and $\boldsymbol{b}$ is the bias. We use the Poisson loss to optimize the representation similarity between artificial neurons (i.e. $\hat{\boldsymbol{s}}$) and biological neurons neurons (i.e. $\boldsymbol{s}$) in the mapping function as Eq.(6),

$$\mathcal{L}(\theta_{\mathrm{s}}, \theta_{\mathrm{d}}) = \sum (\hat{\boldsymbol{s}} - \boldsymbol{s} \log \hat{\boldsymbol{s}}). \qquad (6)$$

Previous studies have shown that the responses of V1 neurons to natural stimuli are sparse, and the activity of neural populations with higher sparseness exhibits greater discrimination against natural stimuli. (Vinje & Gallant, 2000; Weliky et al., 2003; Froudarakis et al., 2014; Yoshida & Ohki, 2020). Likewise, (Zhuang et al., 2017) has reported that the resemblance between the representation of biological neurons and artificial neurons in higher convolutional layers exists only under the sparsity constraint on the CNN, regardless of any other factors (*e.g.*, , model structure, training algorithm, receptive field size, and property of training stimuli). In our study, the representational similarity of V1 neurons is brought to a specific layer of the CAE encoder ($\boldsymbol{h}_i, i \in [1, 2, 3, 4]$) with the sparsity constraint for artificial neurons in this layer.

**CAE-FR:** The CAE-FR combines the function of CAE and CNN-FR. The loss of CAE-FR explicitly considers both the image reconstruction task and the neural representation similarity task, as defined in Eq. equation 7.

$$\mathcal{L}(\phi_1, \phi_2, \theta_{\mathrm{s}}, \theta_{\mathrm{d}}) = \alpha * \|\boldsymbol{x} - f_2(\phi_2, f_1(\phi_1, \boldsymbol{x}))\|_2^2 + \beta * \sum (f_3(\boldsymbol{h}_i, \theta_{\mathrm{s}}, \theta_{\mathrm{d}}) - \boldsymbol{s} \log f_3(\boldsymbol{h}_i, \theta_{\mathrm{s}}, \theta_{\mathrm{d}})). \tag{7}$$

Intuitively, the larger $\alpha$ favors the reconstruction task, while the larger $\beta$ biases toward the neural representation similarity task.

## 4 EXPERIMENTS AND RESULTS

### 4.1 EXPERIMENTAL SETTINGS

**Datasets:** We conduct experiments on a publicly available dataset with the gray-color images as visual stimuli and the corresponding neural responses. The neural response dataset is obtained

from Antolík et al. (2016). The neural responses in the dataset are recorded in three regions in the primary visual cortex (V1) of sedated mice visually stimulated with the natural images (See Appendix Fig. 5). The number of neurons over the three brain regions is shown in the Appendix Table 7.

**Models:** We realized DAE-NRs with nine different combinations of DAE and CNM (See Table 1). DAE-NRs are compared with DAE baselines for the image reconstructions task and compared with CNM baselines for the neural representation similarity task. We choose CAE, VAE and VQ-VAE as the DAE baselines and CNN-FR, CNN-FC and CNN-FM as the CNM baselines, respectively. Especially, we have the CNM with $\mathbf{h}_i$ (CNM$_i$, $i \in \{1, 2, 3, 4\}$). The CNM in DAE-NR can be readout from four different convolutional layers (*i.e.*, $\mathbf{h}_i$ of the DAE encoder), resulting in four variants of DAE-NRs (*i.e.*, DAE-NR$_i$, $i \in \{1, 2, 3, 4\}$) which represent the CNM extracted from the different convolutional layers $\mathbf{h}_i$.

**Network architectures:** The architecture of the DAE in DAE-NR is in the Appendix Table 14. The dimension of latent variable in CAE and VAE is set as 100; the prior of VAE and VQ-VAE is the Gaussian distribution $N(0, 1)$ and uniform distribution, respectively. The dimension of latent embedding space in VQ-VAE is $32 \times 256$. Each convolutional layer is followed by a batch normalization with an ELU activation function, while the activation function in the final layer for image reconstruction is $tanh$. The CNM in DAE-NR shares the four different convolutional layers ($\mathbf{h}_1$, $\mathbf{h}_2$, $\mathbf{h}_3$, and $\mathbf{h}_4$). The CNM in DAE-NR can be readout from four different convolutional layers (i.e. $\mathbf{h}_i$ of the encoder of DAE), so there are four variants of DAE-NR, i.e. DAE-NR$_i$, $i \in \{1, 2, 3, 4\}$ representing the CNM extracted from the different convolutional layers $\mathbf{h}_i$.

**Training procedures:** We preprocess the images (i.e., reshape the size of the natural image to $32 \times 32 \times 1$ and normalize the intensity of the image to [-1,1]) and then input them to the model. The model is trained with an initial learning rate of 0.001, and the early stopping strategy is applied based on a separated validation set. If the error in the validation set does not improve by 1000 steps, we return the best parameter set, reduce the learning rate by two, and train in the second time. The VAE is optimized by the evidence lower bound ($ELBO$), while the VQ-VAE is optimized following the settings in (van den Oord et al., 2017). The settings of hyperparameters for tasks in Sec 4.2 and Sec 4.3 are detailed in Appendix Table 8 and 9, respectively. The Appendix Table 10 and Table 11&12&13 are the settings the hyperparameters of different DAE-NR variants for image reconstruction on Region 3 and neural similarity experiments on Region 1,2 and 3 in Sec 4.4, respectively.

Table 2: The quantitative results of image reconstruction with all neurons in the region 1, 2, and 3, respectively. The best result in each region under different metrics is highlighted with boldface.

| Region | Region 1 | | | Region 2 | | | Region 3 | | |
|---|---|---|---|---|---|---|---|---|---|
| Metric
Model | MSE↓ | PSNR↑ | SSIM↑ | MSE↓ | PSNR↑ | SSIM↑ | MSE↓ | PSNR↑ | SSIM↑ |
| CAE | 0.022 | 23.709 | 0.771 | 0.024 | 23.338 | 0.754 | 0.081 | 17.039 | 0.561 |
| CAE-FR$_1$ | **0.021** | **23.829** | **0.776** | **0.023** | 23.392 | 0.753 | 0.044 | 19.751 | 0.763 |
| CAE-FR$_2$ | **0.021** | 23.779 | 0.775 | **0.023** | 23.440 | 0.759 | **0.043** | **19.819** | **0.764** |
| CAE-FR$_3$ | **0.021** | 23.778 | 0.775 | 0.024 | 23.330 | 0.755 | **0.043** | 19.789 | 0.761 |
| CAE-FR$_4$ | 0.022 | 23.721 | 0.773 | **0.023** | **23.491** | **0.760** | 0.059 | 18.462 | 0.668 |

**Tasks:** There are two tasks in the experiment: 1) the image reconstruction (IR) task and 2) the neural representation similarity (NRS) task. We apply CAE-FR to analyse the effects of neural responses on IR and the effects of image reconstruction on NRS tasks in DAE-NR. We compare the other DAE-NR variants with DAE and CNM for IR and NRS tasks to explore the generalisation capability of DAE-NR. In the IR task, we use the mean squared error (MSE↓), structural similarity (SSIM↑) Wang et al. (2004), and peak signal-to-noise ratio (PSNR↑) Wang et al. (2004) as metrics to quantify the image reconstruction performance[1]. We compare the DAE-NR$_1$, DAE-NR$_2$, DAE-NR$_3$ and DAE-NR$_4$ with the standard DAE. In the NRS task, we use Pearson correlation coefficient (PCC↑) as a metric to quantitatively evaluate models. We implement the traditional end-to-end CNM$_i$ ($i \in \{1, 2, 3, 4\}$) as baseline models.

---

[1]The up arrow ↑ indicates that the higher the value, the better, and the down arrow ↓ indicates that the lower, the better.

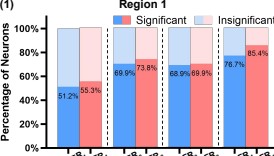 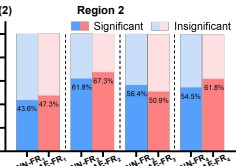 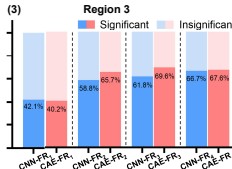

Figure 3: The number of significant neurons (deep blue and deep red) and insignificant neurons (light blue and light red) in region 1, 2, 3 in the image reconstruction task. The threshold for significance is $p \leq 0.05$. The results show that our models (CAE-FRs) provide more significant neurons (red) than the baseline models (blue).

Table 3: The quantitative results of image reconstruction with constraints of significant neurons and insignificant neurons in the region 3. The best result under different metrics is highlighted with boldface.

| Metric | MSE↓ | | SSIM↑ | | PSNR↑ | |
|---|---|---|---|---|---|---|
| Significant? Model | ✓ | × | ✓ | × | ✓ | × |
| CAE-FR$_1$ | **0.043** | 0.125 | **0.761** | 0.332 | **19.784** | 15.168 |
| CAE-FR$_2$ | **0.047** | 0.082 | **0.743** | 0.547 | **19.467** | 16.970 |
| CAE-FR$_3$ | **0.049** | 0.116 | **0.724** | 0.362 | **19.245** | 15.463 |
| CAE-FR$_4$ | 0.047 | **0.045** | 0.740 | **0.752** | 19.497 | **19.628** |

## 4.2 BIOLOGICAL NEURAL RESPONSES IN CAE-FR IMPROVES IMAGE RECONSTRUCTION

Ten examples of reconstructed images by CAE-FR with neural responses in brain region 3 are presented in Fig. 2. It is obvious that CAE-FR models, no matter which layer the neural response is mapped to, can better reconstruct the original images compared with the standard CAE. The results of the brain region 1 and 2 are illustrated in Appendix Fig. 6 and Fig. 7. The quantitative comparisons of image reconstruction performance are listed in Table 2. It shows that CAE-FR$_1$, CAE-FR$_4$, and CAE-FR$_2$ achieve the best performance when the network gets information from the neurons in the regions 1, 2, and 3, respectively. Although our model works in region 3 with a significant improvement, there is a slight improvement in regions 1 & 2. We deem that the neurons carrying more information about essential features of the visual stimulus can reconstruct better. The results in Fig4 and Appendix Fig. 8 show that region 3 has more significant neurons compared to Regions 1 and 2. These results indicate that neurons in regional 3 carry more information than those in regions 1 and 2 about essential features of the visual stimulus.

Fig. 2 and Table 2 suggests that the information from biological neurons could help CAE-FR for image reconstruction. Further questions are *under what circumstances* and *to what extent* CAE-FR is beneficial from the neural response. Our hypothesis is that *only the biological neurons with high representation similarity to artificial neurons can provide information and contribute to CAE-FR*.

To test our hypothesis, we identify the biological neurons that significantly correlate with the artificial neurons in the CAE-FR model on the region 3, as well as the insignificant neurons in each brain region. Fig. 4 shows the number of significant neurons and insignificant neurons in region 3 in the IR task. Then, we train the CAE-FRs with the information from these two groups of neurons in region 3 for the IR task separately. The quantitative results of the three metrics are shown in Table 3, confirming that insignificant neurons carry very little information about the images, and they can not

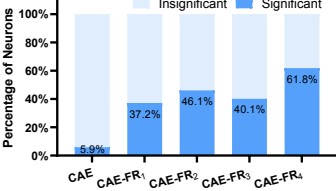

Figure 4: The number of significant neurons and insignificant neurons of the Region 3 in the image reconstruction task. The threshold for significance is $p \leq 0.05$.

Table 4: Pearson correlation between the representations of artificial and biological neurons in region 1, 2, and 3, respectively. The best result in each region under different layers of the model is highlighted with boldface.

| Model / Region | CNN-FR$_1$ | CAE-FR$_1$ | CNN-FR$_2$ | CAE-FR$_2$ | CNN-FR$_3$ | CAE-FR$_3$ | CNN-FR$_4$ | CAE-FR$_4$ |
|---|---|---|---|---|---|---|---|---|
| Region 1 | 0.341 | **0.346** | 0.467 | **0.476** | 0.454 | **0.455** | 0.441 | **0.463** |
| Region 2 | 0.257 | **0.281** | 0.384 | **0.400** | 0.333 | **0.338** | 0.301 | **0.345** |
| Region 3 | 0.246 | **0.260** | 0.361 | **0.388** | 0.372 | **0.404** | 0.385 | **0.393** |

contribute to improving image reconstruction performance. These experiments verify our hypothesis, indicating that information from significant neurons can better guide CAE-FR to reconstruct images.

### 4.3 IMAGE RECONSTRUCTION IN CAE-FR IMPROVES NEURAL REPRESENTATION SIMILARITY

Here, we test image reconstruction's effects on neural representation similarity in CAE-FR. The results of the Pearson correlation coefficient between the neural representation of artificial neurons and of biological neurons in region 1, region 2 and region 3 are shown in Table 4. Our models (*i.e.*, CAE-FR$_i$) obtain larger PCC in all three regions compared to the baseline models without the image reconstruction loss (*i.e.*, CNN-FR$_i$). The results show that image reconstruction in our model favors neural representation similarity. This phenomenon seems counter-intuitive as there is a tradeoff between the IR loss and the NSR loss in Eq. equation 3. We hypothesize that *the benefit may stem from feature learning for image reconstruction by CAE: more biological neurons would share representation similarity with artificial neurons when adding image reconstruction loss*. In fact, our experimental results verify the hypothesis (Fig. 3), suggesting that image reconstruction loss can help CAE-FR to improve neural representation similarity between artificial neurons and biological neurons.

### 4.4 GENERALIZABILITY ACROSS VARIANTS OF CNM AND DAE

To investigate the generalizability of DAE-NR variants, we compare different DAE-NR variants with baseline CNMs and DAEs on IR and NRS tasks, respectively.

**Comparisons in IR task:** Table 5 provides the quantitative comparisons among different models. Specifically, we compare different CNMs variants (*i.e.*, CAE-FR$_i$, CAE-FC$_i$, and CAE-FM$_i$) with the standard CAE on the IR task. The results show that the best performance is achieved when the network leverages information from biological neurons in Region 3. Moreover, to examine generalizability in other DAE variants, we implement some VAE variants (VAE-FR$_i$, VAE-FC$_i$, and VAE-FM$_i$) and VQ-VAE variants (VQ-VAE-FR$_i$, VQ-VAE-FC$_i$, and VQ-VAE-FM$_i$) and compare their performance with the standard VAE and VQ-VAE on the IR task, respectively. Our results show that DAE-NR, VAE variants, and VQ-VAE variants that use information from biological neurons have better IR performances than baseline models.

**Comparisons in NSR task:** Pearson correlation coefficient (PCC↑) between the neural representation of artificial and biological neurons in region 1, region 2, and region 3 are shown in Table 6. To verify the effect of different DAEs for CNN-FR, we compare the variants of CAE (CAE-FR$_i$, CAE-FC$_i$, and CAE-FM$_i$) with CNN-FR on the NSR task. Our models obtain higher PCC in all three regions compared to the baseline models. Furthermore, to validate the function of other CNM variants with different constraints of DAEs, we apply the variants of CNN-FC (CAE-FC$_i$, VAE-FC$_i$, and VQ-VAE-FC$_i$) and the variants of CNN-FM (CAE-FM$_i$, VAE-FM$_i$, and VQ-VAE-FM$_i$) to compare the performance on the NRS task, respectively. Our results show that the biologically-inspired variants of both CNN-FC and CNN-FM via our DAE-NR framework have a better performance than the original models.

Together, we demonstrate that the DAE-NR has a good generalization in variants of CNM and DAE. Adding neural information to guide DAE can help reconstruct visual stimuli; in turn, integrating image reconstruction loss into CNM can improve the representation similarity between artificial and biological neurons.

Table 5: Quantitative results of image reconstruction with all neurons in the region 3. The best results under different metrics for each region are highlighted in bold.

| $\mathbf{h}_i$ Metric Model | $\mathbf{h}_1$ | | | $\mathbf{h}_2$ | | | $\mathbf{h}_3$ | | | $\mathbf{h}_4$ | | |
|---|---|---|---|---|---|---|---|---|---|---|---|---|
| | MSE↓ | PSNR↑ | SSIM↑ | MSE↓ | PSNR↑ | SSIM↑ | MSE↓ | PSNR↑ | SSIM↑ | MSE↓ | PSNR↑ | SSIM↑ |
| $\text{CAE}_i$ | 0.074 | 17.289 | 0.593 | 0.074 | 17.289 | 0.593 | 0.074 | 17.289 | 0.593 | 0.074 | 17.289 | 0.593 |
| $\text{CAE-FR}_i$ **(Ours)** | **0.043** | 19.832 | 0.764 | **0.043** | **19.852** | **0.767** | 0.045 | 19.698 | 0.758 | 0.051 | 19.107 | 0.718 |
| $\text{CAE-FC}_i$ **(Ours)** | **0.043** | **19.876** | **0.768** | 0.044 | 19.811 | 0.763 | 0.061 | 18.290 | 0.734 | **0.044** | **19.802** | **0.760** |
| $\text{CAE-FM}_i$ **(Ours)** | 0.045 | 19.692 | 0.761 | **0.043** | 19.843 | 0.764 | **0.044** | **19.818** | **0.761** | 0.044 | 19.757 | 0.757 |
| $\text{VAE}_i$ | 0.104 | 15.983 | 0.402 | 0.104 | 15.983 | 0.402 | 0.104 | 15.983 | 0.402 | 0.104 | 15.983 | 0.402 |
| $\text{VAE-FR}_i$ **(Ours)** | **0.101** | **16.106** | 0.414 | **0.101** | 16.066 | 0.409 | **0.101** | 16.083 | **0.417** | 0.102 | 16.063 | 0.414 |
| $\text{VAE-FC}_i$ **(Ours)** | 0.103 | 16.003 | 0.412 | **0.101** | **16.094** | 0.412 | **0.101** | **16.096** | 0.416 | **0.101** | **16.089** | **0.418** |
| $\text{VAE-FM}_i$ **(Ours)** | 0.102 | 16.046 | **0.415** | **0.101** | 16.073 | **0.414** | 0.102 | 16.025 | 0.413 | 0.103 | 16.017 | 0.412 |
| $\text{VQ-VAE}_i$ | 0.082 | 16.959 | 0.497 | 0.082 | 16.959 | 0.497 | 0.082 | 16.959 | 0.497 | 0.082 | 16.959 | 0.497 |
| $\text{VQ-VAE-FR}_i$ **(Ours)** | **0.071** | 17.636 | 0.569 | 0.069 | 17.741 | 0.568 | 0.073 | 17.487 | **0.557** | 0.071 | 17.581 | 0.560 |
| $\text{VQ-VAE-FC}_i$ **(Ours)** | 0.076 | 17.264 | 0.526 | 0.075 | 17.324 | 0.525 | **0.072** | **17.539** | 0.547 | **0.067** | **17.839** | **0.581** |
| $\text{VQ-VAE-FM}_i$ **(Ours)** | 0.068 | **17.782** | **0.572** | **0.068** | **17.791** | **0.571** | 0.091 | 16.496 | 0.546 | 0.072 | 17.508 | 0.542 |

Table 6: Results of PCC↑ between the representations of artificial and biological neurons in three brain regions. The best results are highlighted in bold.

| Region $\mathbf{h}_i$ Model | Region 1 | | | | Region 2 | | | | Region 3 | | | |
|---|---|---|---|---|---|---|---|---|---|---|---|---|
| | $\mathbf{h}_1$ | $\mathbf{h}_2$ | $\mathbf{h}_3$ | $\mathbf{h}_4$ | $\mathbf{h}_1$ | $\mathbf{h}_2$ | $\mathbf{h}_3$ | $\mathbf{h}_4$ | $\mathbf{h}_1$ | $\mathbf{h}_2$ | $\mathbf{h}_3$ | $\mathbf{h}_4$ |
| $\text{CNN-FR}_i$ | 0.335 | 0.455 | 0.432 | 0.431 | 0.254 | 0.375 | 0.309 | 0.305 | 0.244 | 0.348 | 0.350 | 0.359 |
| $\text{CAE-FR}_i$ **(Ours)** | 0.338 | 0.471 | 0.449 | **0.452** | 0.277 | **0.396** | 0.319 | 0.348 | 0.254 | 0.377 | **0.397** | 0.394 |
| $\text{VAE-FR}_i$ **(Ours)** | 0.338 | **0.479** | 0.455 | 0.450 | 0.277 | 0.386 | 0.317 | 0.342 | **0.269** | 0.375 | 0.382 | **0.404** |
| $\text{VQ-VAE-FR}_i$ **(Ours)** | **0.349** | 0.470 | **0.463** | 0.444 | **0.295** | 0.388 | **0.337** | 0.349 | 0.261 | **0.394** | 0.382 | 0.387 |
| $\text{CNN-FC}_i$ | 0.343 | 0.349 | 0.397 | 0.435 | 0.247 | 0.268 | 0.292 | 0.328 | 0.325 | 0.316 | 0.344 | 0.374 |
| $\text{CAE-FC}_i$ **(Ours)** | **0.349** | 0.382 | 0.415 | **0.438** | **0.266** | 0.281 | 0.319 | 0.341 | 0.335 | **0.341** | 0.384 | 0.399 |
| $\text{VAE-FC}_i$ **(Ours)** | 0.340 | **0.383** | 0.418 | **0.438** | 0.255 | **0.290** | 0.321 | 0.343 | 0.337 | 0.337 | 0.379 | 0.398 |
| $\text{VQ-VAE-FC}_i$ **(Ours)** | 0.347 | 0.377 | 0.415 | 0.434 | 0.264 | 0.289 | 0.313 | **0.361** | **0.338** | **0.341** | **0.388** | **0.427** |
| $\text{CNN-FM}_i$ | 0.150 | 0.222 | 0.110 | 0.220 | 0.123 | 0.186 | 0.127 | 0.162 | 0.064 | 0.140 | 0.102 | 0.178 |
| $\text{CAE-FM}_i$ **(Ours)** | **0.154** | 0.235 | 0.115 | 0.227 | 0.134 | 0.222 | **0.144** | **0.182** | 0.073 | 0.171 | 0.109 | **0.203** |
| $\text{VAE-FM}_i$ **(Ours)** | 0.149 | 0.238 | **0.125** | **0.240** | **0.136** | **0.224** | 0.131 | 0.176 | **0.074** | **0.172** | **0.111** | 0.201 |
| $\text{VQ-VAE-FM}_i$ **(Ours)** | 0.153 | **0.239** | 0.115 | 0.221 | 0.135 | 0.221 | 0.136 | 0.162 | 0.069 | 0.162 | **0.111** | 0.200 |

## 5 CONCLUSION

In this study, we proposed the DAE-NRs, a hybrid framework that integrates the neural response into deep autoencoder models. Inspired by CNN-based neural activity prediction models in computational neuroscience Klindt et al. (2017); Bashivan et al. (2019), we leveraged a Poisson loss to bring neural information to a specific layer of DAE, resulting in better image reconstruction performance. In turn, the IR task facilitates feature learning in DAE and leads to higher neural representation similarity between biological and artificial neurons. Our work bridges the gap between DAE for image reconstruction and CNM for neural representation similarity.

**Broader impact** Besides the image reconstruction task and neural representation similarity task, DAE-NR could enable many other potential applications in the future. For instance, DAE-NR provides a more natural way to synthesize images that maximize or control neural activity, compared with the method proposed by Bashivan et al. (2019). Also, DAE-NR can serve as a data engine for evirtual experiments, such as synthesizing neural responses to visual stimuli. Moreover, it is important to investigate the generalizability of DAE-NR on variants of auto-encoder Larsen et al. (2016); Khattar et al. (2019); Park et al. (2020); Wang et al. (2021); Cai et al. (2021), on various types of stimuli (*e.g.*, sound and face) and in other tasks (*e.g.*, classification, generation and detection Ran et al. (2021)). Although we only tested DAE-NR with mice neural data, the DAE-NR framework can be easily extended to the neurons in primates (*e.g.*, monkey Zhang et al. (2021) or humans). DAE-NR opens a new window for combining artificial intelligence and brain intelligence.

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
