# OpenReview forum: "A computational framework to unify representation similarity and function in biological and artificial neural networks"
_ICLR.cc/2023/Conference — Submitted to ICLR 2023_

### Official Review · Reviewer_n7th · 2022-10-23

**Confidence:** 4
**Correctness:** 2
**Technical Novelty And Significance:** 2
**Empirical Novelty And Significance:** 1
**Recommendation:** 3

**Clarity, Quality, Novelty And Reproducibility:**

The paper is written in a reasonably clear way, and the idea of
regularizing a predictor of neural encoding with a VAE is
interesting. However, the quality of the work is not satisfactory in
my view, because - as detailed above - several of the claims made by
this work are not supported by the evidence. Regarding
reproducibility, the methods are described reasonably well, but I
could not find a code link.

**Strength And Weaknesses:**

## Strengths
1. The idea of using a VAE as a regularizer for a neural activity
   prediction task is interesting and (to my knowledge) novel, and
   executed in a clean way. The methods and results of the research
   are reasonably well described in the paper (with a few minor
   exception, noted below).
2. The claim that inclusion of the VAE improves representational
   similarity is backed up by the data, if it is interpreted in the
   narrow sense of only being related to the specific architecture at
   hand. However, this comparison is made in complete isolation from
   the existing literature on representational similarity (see point 3
   below).

## Weaknesses
1. It is not clear what is meant by the "image reconstruction
   problem", and how can the proposed architecture bring progress on
   that. In the introduction, it is stated that the image
   reconstruction problem is an important one in computer vision, and
   a couple of references are given to papers by J Fessler on medical
   applications. Is there any way in which DAE-NR could contribute to
   this type of applications?
2. In the abstract and the introduction, the "guidance of neural
   information" is offered as a general principle that should somehow
   improve image reconstruction performance. However, in the present
   work, this is only shown to be true for the neural data from one of
   the three regions recorded in  Antonlik et al
   2016 (region 3). Including
   data from the other recorded regions (regions 1 and 2) did not
   improve appreciably the performance of the autoencoder (table 2),
   invalidating the generality of the approach, but no comment or
   justification is proposed for this fact. What makes neurons in
   region 3 special? Is there a theoretical or practical reason why
   inclusion of the data for these neurons, but not the others, should
   help the autoencoder? Or is it simply that these neurons happen to
   carry a bit more information than those in regions 1 and 2 about
   important features of the visual stimulus, and therefore their
   inclusion helps learning better representations in the autoencoder?
3. For neural activity prediction, no comparison is made with the
   state of the art (only comparisons with variants of the proposed
   architecture are done). This is particularly surprising given that
   the neural data used in this paper comes from another work where
   neural prediction was performed. In Figure 5A of Antonlik et al
   2016, the correlation coefficient between model prediction and
   neural activity is shown to be about 0.5 for all three regions. In
   the present work (Table 4), this metric is somewhat lower (a range
   of about 0.3-0.5, considering 8 different models).
4. The analysis of "significant neurons" in table 3 seem
   overinterpreted. If I've understood correctly the procedure for
   this analysis, it is as follows: after training the network,
   neurons (only those for region 3, in the main text) are divided
   among those for which the model prediction are better than chance
   (the "significant" ones), and those for which they aren't. Then the
   network is re-trained, from scratch, using only data from either
   significant or insignificant neurons. It is observed that the
   significant neurons contribute to improving the VAE, while the
   insignificant ones don't. It is claimed that "These experiments
   verify our hypothesis, indicating that information from significant
   neurons can guide CAE-FR to better reconstruct images". I am not
   sure what is added by this analysis: the simplest explanation is
   that the insignificant neurons don't carry any (or carry very
   little) information about the images, so they are effectively
   random noise generators for the purpose of image reconstruction,
   and therefore can't possibly contribute to improving it. In other
   words, in my opinion this analysis is just showing in a different
   way that some neurons carry some information about the images, and
   some don't. In this sense, saying that information from informative
   neurons can contribute to a better reconstruction is somewhat of a
   vacuous claim; for instance, one could do even better by including
   skip connections directly from the image input layer.
5. Additionally, the numbers in the "significant" column in Table 3
   seem uniformly worse than the numbers in Table 2. In other words,
   it seems that training with the significant neurons alone is better
   than training with the insignificant neurons alone, but worse than
   training with all neurons. This seems relevant and should be
   discussed. How do the authors square this fact with the statement
   that "insignificant neurons jeopardize CAE-FR"?
6. Why are the images in Figure 2 so small and featureless? The
   abstract states that the input images are "natural images", and the
   paper says that the data is taken from Antonlik et al 2016. But the
   images that paper (see for instance Figure 1 in Antonlik et al)
   look completely different, and the methods section of that paper
   states "The stimulus set was composed of static scenes from David
   Attenborough’s BBC documentary Life of Mammals, depicting natural
   scenes such as landscapes, animals or humans. Images were scaled to
   have 256 equally spaced luminance steps, and were composed of
   384×208 pixels, and expanded to fill the screen". I must be missing
   something, because the images in Figure 2 look like random blobs to
   me.

**Summary Of The Paper:**

This paper presents a deep neural network that jointly learns 1) to
encode images as a variational autoencoder, and 2) to use the images
to predict neuronal activities recorded in mouse visual cortex (in
response to the same set of images, presented as visual stimuli). By
sharing the representations used to perform both tasks, the goal of is
to improve the neural prediction performance by using the autoencoder
as a regularizer, and, reciprocally, to improve the autoencoder's
reconstruction quality by making its internals more similar to
putative processes of biological vision.

**Summary Of The Review:**

In my opinion, this paper fails to back up some of its core claims and
to justify the (claimed) relevance for the proposed architecture from
a computer vision standpoint. Therefore, my recommendation is not to
accept this work for publication.

---

> ### Author Response · Authors · 2022-11-18
> **Response to Reviewer n7th: Part 1**
>
> **Q1: It is not clear what is meant by the "image reconstruction problem", and how can the proposed architecture bring progress on that. In the introduction, it is stated that the image reconstruction problem is an important one in computer vision, and a couple of references are given to papers by J Fessler on medical applications. Is there any way in which DAE-NR could contribute to this type of applications?**
>
> > **Response to Q1 :** Thank you for your good questions.
>
> * > **It is not clear what is meant by the “image reconstruction problem”, and how can the proposed architecture bring progress on that.**
> We apologize for the confusion about ‘image reconstruction problem’. The image reconstruction problem here means to reconstruct the input images by learning the latent representation using deep auto-encoders (DAEs). The prior of DAEs is usually set as a Gaussian prior for VAE and a uniform prior for VQVAE. The setting of prior would largely influence the image reconstruction performance on DAEs[1]. In our work, by adding neural information as the prior knowledge to guide the learning process of DAE, the proposed architecture can help reconstruct the visual stimuli.
>
> [1] Jakub Tomczak and Max Welling. Vae with a vampprior. In International Conference on Artificial Intelligence and Statistics, pp. 1214–1223. PMLR, 2018.
>
>
> * > **In the introduction, it is stated that the image reconstruction problem is an important one in computer vision, and a couple of references are given to papers by J Fessler on medical applications**.
> Although DAE can work well in solving medical imaging reconstruction [2], we do not specifically validate our method on medical imaging applications. Image reconstruction is important in medical applications, which help dimensionality reduction, feature extraction and data generation. That is the reason why we cited some references in medical applications. To address your comments, we have revised the sentence as “Image reconstruction is one of the essential tasks [2][3][4].”
> * > **Is there any way in which DAE-NR could contribute to this type of applications?**
> There are two potential ways that DAE-NR could contribute to medical applications. one is to add other available information about the medical image, which property is the same as the neural response,  to guide the medical image reconstruction task. Another one is to simulate the fake neural response of the image (e.g., medical image, natural image) and then add this fake neural response to guide the model to reconstruct the original image.
>
> [2] Saiprasad Ravishankar, J. C. Ye, and Jeffrey A. Fessler. Image reconstruction: From sparsity to data-adaptive
> methods and machine learning. Proceedings of the IEEE, 108:86–109, 2020.
>
> [3]Hinton, G. E., & Salakhutdinov, R. R. (2006). Reducing the dimensionality of data with neural networks. science, 313(5786), 504-507.
>
> [4]Kingma, D. P., & Welling, M. (2013). Auto-encoding variational bayes. arXiv preprint arXiv:1312.6114.

---

> > ### Author Response · Authors · 2022-11-18
> > **Response to Reviewer n7th: Part 2**
> >
> > **Q2: In the abstract and the introduction, the "guidance of neural information" is offered as a general principle that should somehow improve image reconstruction performance. However, in the present work, this is only shown to be true for the neural data from one of the three regions recorded in Antonlik et al 2016 (region 3). Including data from the other recorded regions (regions 1 and 2) did not improve appreciably the performance of the autoencoder (table 2), invalidating the generality of the approach, but no comment or justification is proposed for this fact. What makes neurons in region 3 special? Is there a theoretical or practical reason why inclusion of the data for these neurons, but not the others, should help the autoencoder? Or is it simply that these neurons happen to carry a bit more information than those in regions 1 and 2 about important features of the visual stimulus, and therefore their inclusion helps learning better representations in the autoencoder?**
> >
> > > **Response to Q2 :** Thank you for your good questions.
> >
> > * >  **In the abstract and the introduction, the "guidance of neural information" is offered as a general principle that should somehow improve image reconstruction performance. However, in the present work, this is only shown to be true for the neural data from one of the three regions recorded in Antonlik et al 2016 (region 3). Including data from the other recorded regions (regions 1 and 2) did not improve appreciably the performance of the autoencoder (table 2), invalidating the generality of the approach, but no comment or justification is proposed for this fact.**
> > We are sorry for the misleading. Our model is working well in all three regions, with a great improvement in region 3 and a slight improvement in regions 1 & 2. **DAE-NR can improve the image reconstruction quality with the help of a Poisson loss on the predicted neural activity and the guidance of neural information.** We have claimed this point in the last paragraph of the introduction (**please see it in the following text**). Moreover, we describe the framework of DAE-NR in the method (See 3.1), and the biological neurons with high representation similarity to artificial neurons can provide information and contribute to CAE-FR in the experiment (See 4.2)
> >
> > *"We propose a united biologically inspired framework, which jointly learns i) to project features of visual input in a specific layer of the encoder to biological neural responses by a mapping function and ii) to reconstruct the visual input via the decoder."*
> > *"The deep auto-encoders embedded in DAE-NR can improve the image reconstruction quality with the help of a Poisson loss on the predicted neural activity, compared to the baseline auto-encoder models (Sec.4.2 & 4.4)."*
> >
> > * > **What makes neurons in region 3 special? Is there a theoretical or practical reason why inclusion of the data for these neurons, but not the others, should help the autoencoder? Or is it simply that these neurons happen to carry a bit more information than those in regions 1 and 2 about important features of the visual stimulus, and therefore their inclusion helps learning better representations in the autoencoder?**
> > The results in Fig4 and  Appendix Fig. 8 show that region 3 has more significant neurons, compared to Regions 1 and 2. These results indicate that neurons in regional 3 carry more information than those in regions 1 and 2 about essential features of the visual stimulus. And the neuron in region 3 can guide the latent space of DAE to better encode information of the input image. Therefore their inclusion helps to learn better representations in the CAE-FR.

---

> > > ### Author Response · Authors · 2022-11-18
> > > **Response to Reviewer n7th: Part 3**
> > >
> > > **Q3: For neural activity prediction, no comparison is made with the state of the art (only comparisons with variants of the proposed architecture are done). This is particularly surprising given that the neural data used in this paper comes from another work where neural prediction was performed. In Figure 5A of Antonlik et al 2016, the correlation coefficient between model prediction and neural activity is shown to be about 0.5 for all three regions. In the present work (Table 4), this metric is somewhat lower (a range of about 0.3-0.5, considering 8 different models).**
> > >
> > > > **Response to Q3**:  Thanks for your good questions.
> > > >
> > > * > **For neural activity prediction, no comparison is made with the state of the art (only comparisons with variants of the proposed architecture are done)**.
> > > We compared the **three CNM baselines model** (e.g., CNN-FR$_i$ (SOTA), CNN-FC$_i$ and CNN-FM$_i$), as well as other variants, for the neural activity prediction performance. Please see the comparison results shown in **Tables 4** and **6**.
> > > * > **This is particularly surprising given that the neural data used in this paper comes from another work where neural prediction was performed. In Figure 5A of Antonlik et al 2016, the correlation coefficient between model prediction and neural activity is shown to be about 0.5 for all three regions. In the present work (Table 4), this metric is somewhat lower (a range of about 0.3-0.5, considering 8 different models).**
> > > We changed the scale of the original input shape (31, 31, 1)  of visual stimuli in  Antonlik et al. 2016[5]  to (32, 32, 1) in our work. We normalized the original visual stimuli to (-1, 1) to use the tanh activation function for reconstruction. The HSM model in Antonlik et al. 2016 is similar to CNN-FRi in table 1 of Klindt et al.  2017[6].  It is not a CNM based on the CNN model. The architecture and the loss function of our model differ from the models in Klindt et al. 2017. We set the different architectures to prove the generalization of our model and to trade off the image reconstruction and neural similarity tasks. The original version of CAE-FR, CAE-FM, and CAE-FC are trained by the smoothness, group sparsity and Poisson loss on the neural prediction tasks. We only keep the two loss terms (the passion loss for neural similarity and the mse loss for image reconstruction ) in our work in order to reduce the influence of other losses. **The performance is lower than other SOTA models of  Antonlik et al. 2016 in Tables 4 and 6 because we validate the impact of image reconstruction for the neural prediction task.**
> > >
> > > [5]Ján Antolík, S. Hofer, J. Bednar, and T. Mrsic-Flogel. Model constrained by visual hierarchy improves prediction of neural responses to natural scenes. PLoS Computational Biology, 12, 2016.
> > > [6]David A Klindt, Alexander S Ecker, Thomas Euler, and Matthias Bethge. Neural system identification for large populations separating what and where. In Advances in Neural Information Processing Systems (NeurIPS), pp. 3509–3519, 2017.

---

> > > > ### Author Response · Authors · 2022-11-18
> > > > **Response to Reviewer n7th: Part 4**
> > > >
> > > > **Q4: The analysis of "significant neurons" in table 3 seem overinterpreted. If I've understood correctly the procedure for this analysis, it is as follows: after training the network, neurons (only those for region 3, in the main text) are divided among those for which the model prediction are better than chance (the "significant" ones), and those for which they aren't. Then the network is re-trained, from scratch, using only data from either significant or insignificant neurons. It is observed that the significant neurons contribute to improving the VAE, while the insignificant ones don't. It is claimed that "These experiments verify our hypothesis, indicating that information from significant neurons can guide CAE-FR to better reconstruct images". I am not sure what is added by this analysis: the simplest explanation is that the insignificant neurons don't carry any (or carry very little) information about the images, so they are effectively random noise generators for the purpose of image reconstruction, and therefore can't possibly contribute to improving it. In other words, in my opinion this analysis is just showing in a different way that some neurons carry some information about the images, and some don't. In this sense, saying that information from informative neurons can contribute to a better reconstruction is somewhat of a vacuous claim; for instance, one could do even better by including skip connections directly from the image input layer.**
> > > >
> > > > > **Response to Q4:** Thanks for your good questions.
> > > > * > **The analysis of “significant neurons” in table 3 seem overinterpreted.**
> > > > We agree with reviewer’s **opinion** that our analysis shows in a different way that some neurons carry some information about the images, and some don't. From this point, this analysis in table 3 can support our hypothesis that information from significant neurons which carry some information about the images can guide CAE-FR to reconstruct images better.
> > > > * > **In this sense, saying that information from informative neurons can contribute to a better reconstruction is somewhat of a vacuous claim; for instance, one could do even better by including skip connections directly from the image input layer.**
> > > > Skipping connections directly from the image input layer or previous layer (Res-Net) can obtain good performance on some tasks. But this feature of the previous layer or image input layer does not increase any other modal information about the image. The neural information with some unique pattern can provide additional modal information that does not differ from the neural network or the image. **That's why** significant neurons can guide CAE-FR to reconstruct images better and this analysis **is a necessity**.
> > > >
> > > > **Q5: Additionally, the numbers in the "significant" column in Table 3 seem uniformly worse than the numbers in Table 2. In other words, it seems that training with the significant neurons alone is better than training with the insignificant neurons alone, but worse than training with all neurons. This seems relevant and should be discussed. How do the authors square this fact with the statement that "insignificant neurons jeopardize CAE-FR"?**
> > > >
> > > > > **Response to Q5: Thanks for your good questions.**
> > > > * >**the numbers in the “significant” column in Table 3 seem uniformly worse than the numbers in Table 2.**
> > > > CAE-FR1 and CAE-FR4 in the "significant" column in table 3 are better than in table 2 and CAE-FR2 and CAE-FR3 in the "significant" column in table 3 are slightly worse than in table 2. The training with all neurons is similar to the training with the significant neurons indicating that the neural information is very robust and can reduce the influence of insignificant neurons.
> > > > * > **This seems relevant and should be discussed. How do the authors square this fact with the statement that “insignificant neurons jeopardize CAE-FR”?**
> > > > We **agree with the reviewer's Q4 statement** that insignificant neurons do not carry any (or carry very little) information about the images, so they are effectively random noise generators for image reconstruction and, therefore, can not possibly contribute to improving it. **We fix our statement** in the following:
> > > >
> > > > *"Insignificant neurons carry very little information about the images, and they can not contribute to improving image reconstruction performance. "*

---

> > > > > ### Author Response · Authors · 2022-11-18
> > > > > **Response to Reviewer n7th: Part 5**
> > > > >
> > > > > **Q6: Why are the images in Figure 2 so small and featureless? The abstract states that the input images are "natural images", and the paper says that the data is taken from Antonlik et al 2016. But the images that paper (see for instance Figure 1 in Antonlik et al) look completely different, and the methods section of that paper states "The stimulus set was composed of static scenes from David Attenborough’s BBC documentary Life of Mammals, depicting natural scenes such as landscapes, animals or humans. Images were scaled to have 256 equally spaced luminance steps, and were composed of 384×208 pixels, and expanded to fill the screen". I must be missing something, because the images in Figure 2 look like random blobs to me.**
> > > > >
> > > > > > **Response to Q6: Thanks for your good questions.**
> > > > > * >**Why are the images in Figure 2 so small and featureless?**
> > > > > The images were constrained to this region of interest and then down-sampled to 31×31 pixels to form the input stimuli set. Please see the last sentence in the “ Stimulus presentation protocol and data pre-processing “ paragraph of Antonlik et al. 2016[5]. This data pre-processing is in the following:
> > > > >
> > > > > *“The images were constrained to this region of interest and then down-sampled to 31×31 pixels to form the input stimuli set, which was used in all the subsequent analysis.”*
> > > > >
> > > > > * >  you can download this data from: [ https://figshare.com/articles/dataset/Model_Constrained_by_Visual_Hierarchy_Improves_Prediction_of_Neural_Responses_to_Natural_Scenes/3923658 ] And you can also see the down-sampled image on the **Appendix Figure 5**.

---

### Official Review · Reviewer_YdJC · 2022-10-24

**Confidence:** 4
**Clarity, Quality, Novelty And Reproducibility:** clear, novel and the quality seems good.
**Correctness:** 4
**Technical Novelty And Significance:** 3
**Empirical Novelty And Significance:** 4
**Recommendation:** 8

**Strength And Weaknesses:**

Strength: Transfer learning based on latent features of a CAE for neural data fitting is not new.  The finding that introducing a loss to maximize the representational similarity of the latent feature representations in a CAE with the real neural activities can improve the image reconstruction performance of an autoencoder is novel and interesting.


**Summary Of The Paper:**

This paper shows that optimizing a CAE model to fit neural data and reconstruct input images at the same time, actually improves the performance of both tasks.  The finding that integrating neural response into deep autoencoder models can improve its performance is quite interesting and novel.

**Summary Of The Review:**

This is a relatively simple paper that shows optimizing a CAE model to fit neural data and reconstruct input images simultaneously can actually improves the performance of both tasks.  The approach of integrating neural fitting with CAE for image reconstruction is novel and that fact that it actually shows improvement in performance is reasonable but still a bit surprising, and hence interesting.

---

> ### Author Response · Authors · 2022-11-18
> **Response to Reviewer YdJC**
>
>
>
> Thanks for your comments that this paper that introducing a loss to maximize the representational similarity of the latent feature representations in a CAE with the real neural activities can improve the image reconstruction performance of an autoencoder is **novel and interesting** and this paper is **clear, novel and the quality good**.

---

### Official Review · Reviewer_bubd · 2022-10-25

**Confidence:** 2
**Correctness:** 3
**Technical Novelty And Significance:** 3
**Empirical Novelty And Significance:** 2
**Recommendation:** 5

**Clarity, Quality, Novelty And Reproducibility:**

I found the paper a bit difficult to follow (see miscellaneous comments above).

I do not see indication of a public codebase.

**Strength And Weaknesses:**

(strengths)

It is an interesting idea to merge a CAE and a Convolutional neural model (CNM).

(weaknesses)
I am troubled by the basic assumption that the neurons in mouse V1, V2, etc can be mapped to layers in a CNN.
It is not clear if the gains in image reconstruction are due to the inner layer - Vi fittings, or to extra parameters or the imposed sparsity.

Miscellaneous:

bottom pg 1: Are the structural and functional relationships between ANNs and BNNs sufficiently deep to build on, or are they superficial?

Page 2 "however, hot to ... question": This sets up an expectation. Perhaps remove it, or say that you address this question.

Fig 1: I believe that layers in a CNN (CAE) do not map 1-to-1 to visual layers in mice. Rather, previous studies showed that given a DNN with M layers, you can map a subset of those layers to Vi

Fig 2: why are the CAE images so fuzzy? Doesn't it give good image reconstruction? This is counter-intuitive, unless it is way too small a network, which would suggest that the improved results from an added CNM + fitting are due to extra parameters.

Fig 3: Are you able to add error bars to see if these differences are big? Also, is a higher number of significant neurons better? I believe that in Vi, given a constrained set of images, several neurons might be relatively uninvolved, so higher might be inaccurate. (I don't know).

Typos: there are various repeated words (eg "however however") , and also a repeated comma.

Page 4: "turns to be" -> "turns out to be"

Bottom of page 4: add sub-heading after description of CNM to clarify that the topic is now the hybrid DAE-NR.

eqn 3: beta = 1- alpha?

Abbreviations could be made clearer, noted at time of first introduction. An example is CNM, which I thought was a typo for most of the paper. Also, what is VQ?

Please note when figures and tables (eg fig 5) are in the Appendix.

"Network architectures": Maybe move the architecture description to the appendix where it can be more complete. The 8C88-9CD77 notation is confusing and leaves out detail.

Also: what does the "32 x 256" dimension for VQ-VAE relate to?

page 5" by applying a sparse mask". the explanation of why this is done comes much later. Could you move that explanation so that it follows directly (or precedes)?

Table 2 and 5: How do you restrict the CAE image reconstruction to use only a single layer? This is not described in 3.2.

Table 5: do you have error bars (std devs)? Related: what is the appropriate number of decimal places?



**Summary Of The Paper:**

The paper builds a NN to match (a) image reconstruction and (b) neural similarity of the inner layers of the CNN to neural representations (NR) in a mouse visual cortex. The loss function minimizes image error and NR error.

The results assert that the NN does much better than a convolutional autoencoder (CAE) on image reconstruction, and also (as expected by construction) on NR.

The basic notion (if I got it right) is that if you fit the encoding layers of a CAE to neural representations from a mouse visual cortex, the CAE will have better image reconstructions. It's unclear to me is whether this effect (reported in the paper) is due to the extra parameters provided by the attached CNM, or by the sparsity prior on model weights, rather than to the NR matching. If it is due strictly to the NR matching, that would be, I think, a big thing.

**Summary Of The Review:**

I am perhaps the wrong person to review this work, since I am not confident that the suggested parallels between CNNs and BNNs,  which motivate this work, are more than superficial (eg, the main layers can be made to somewhat correspond). Also, I am not up on this literature. This poor fit is reflected in my Confidence score and other scores.

The accumulation of non-clarities (see "miscellaneous" items above), combined with my skepticism about the extent of the CNN-BNN mapping, make me less favorable to the work. But I emphasize my lack of confidence in this assessment.

---

> ### Author Response · Authors · 2022-11-18
> **Response to Reviewer bubd: Part 1**
>
> Thanks for your comments that this paper is an **interesting idea** to merge a CAE and a Convolutional neural model (CNM). We will **update the code** in the **supplementary** files. We have made revisions to our paper based on your feedback and addressed your concerns in more detail in the miscellaneous below.
>
> **W1: I am troubled by the basic assumption that the neurons in mouse V1, V2, etc can be mapped to layers in a CNN. It is not clear if the gains in image reconstruction are due to the inner layer - Vi fittings, or to extra parameters or the imposed sparsity.**
>
> > **Response to w1**:  Thank you for your comments.
> * > **I am troubled by the basic assumption that the neurons in mouse V1, V2, etc can be mapped to layers in a CNN.**
> According to work in visual systems neuroscience, cortical regions in the brain are hierarchically organized to make invariant object recognition behavior. Several people have built biological inspirations from it and later generalized them to a more general class of computational architectures called Hierarchical Convolutional Neural Networks (HCNNs) [1,2,3,4]. Based on the above research, we are continuously studying the relationship between HCNN and neurons in mouse V1, V2 and etc.
> * > **It is not clear if the gains in image reconstruction are due to the inner layer - Vi fittings, or to extra parameters or the imposed sparsity.**
> The experiments show that neural information from neurons v1, which carry some robust local features about the images, can guide CAE-FR to reconstruct images better. These gains in image reconstruction are due to the inner layer – v1 fitting, and the constraints of image reconstruction loss can make a good performance on neural representation similarity.
>   >
>   > The CNM part of our model has a few parameters  in the linear mapping function that almost can not increase the performance of image reconstruction and do not directly make the image reconstruction better.
>   >
>   > The image reconstruction task needs to learn the local detail features of the image to reconstruct the original images. The imposed sparsity will lose the detailed local features, leading to worse performance on image reconstruction.
>
> [1] LeCun, Y., & Bengio, Y. (1995). Convolutional networks for images, speech, and time series. The handbook of brain theory and neural networks, 3361(10), 1995.
> [2] Yamins, D. L., & DiCarlo, J. J. (2016). Using goal-driven deep learning models to understand sensory cortex. Nature neuroscience, 19(3), 356-365.
> [3] P. Bashivan, Kohitij Kar, and J. DiCarlo. Neural population control via deep image synthesis. Science, 364, 2019.
> [4] Walker, Edgar Y., et al. "Inception loops discover what excites neurons most using deep predictive models." Nature neuroscience 22.12 (2019): 2060-2065.
>
> **Miscellaneous:**
>
> **Q1: bottom pg 1: Are the structural and functional relationships between ANNs and BNNs sufficiently deep to build on, or are they superficial?**
>
> > **Response to Q1: Thank you for your valuable comments.**
> >
> > The fields of neuroscience and artificial intelligence have a long and intertwined history [5]. In recent times, as you say, the **relationships between ANNs and BNNs have been superficial as both subjects have grown immensely in complexity,** and disciplinary boundaries have become more established. However, exploring their correlation is still urgently needed [6]. Moreover, the **hierarchical structure and the functional object recognition  of  CNNs  is very similar to the primary visual cortex** [2,3]
>
> [5] Hassabis, D., Kumaran, D., Summerfield, C., & Botvinick, M. (2017). Neuroscience-inspired artificial intelligence. Neuron, 95(2), 245-258.
> [6] Zador, A., Richards, B., Ölveczky, B., Escola, S., Bengio, Y., Boahen, K., ... & Tsao, D. (2022). Toward Next-Generation Artificial Intelligence: Catalyzing the NeuroAI Revolution. arXiv preprint arXiv:2210.08340.
>
> **Q2: Page 2 "however, how to ... question": This sets up an expectation. Perhaps remove it, or say that you address this question.**
>
>
> > **Response to Q2**:  Thank you for your valuable comments. We have revised the manuscript according to your suggestions. We remove this sentence from our manuscript.
> >
> *"How to build artificial neural networks most similar to biological neural representations remain an open question."*

---

> ### Author Response · Authors · 2022-11-18
> **Response to Reviewer bubd: part 2**
>
> **Q3: Fig 1: I believe that layers in a CNN (CAE) do not map 1-to-1 to visual layers in mice. Rather, previous studies showed that given a DNN with M layers, you can map a subset of those layers to Vi**
>
> > **Response to Q3:  Thank you for your valuable comments.** We follow the previous study[7,8]. The CNM model (CNN-FR, CNN-FC, CNN-FM) usually consist of the encoder and the readout layer. The encoder can be used to extract the image’s features. The readout layer is used to map the feature space H$_i$ to the neural responses in the space. In a future study, we will map a subset of those layers to Vi for other tasks.
>
>
> [7] David A Klindt, Alexander S Ecker, Thomas Euler, and Matthias Bethge. Neural system identification for large populations separating what and where. In Advances in Neural Information Processing Systems (NeurIPS), pp. 3509–3519, 2017.
> [8] Lane McIntosh, Niru Maheswaranathan, Aran Nayebi, Surya Ganguli, and Stephen Baccus. Deep learning models of the retinal response to natural scenes. Advances in neural information processing systems(NeurIPS), 29:1369–1377, 2016
>
>
> **Q4: Fig 2: why are the CAE images so fuzzy? Doesn't it give good image reconstruction? This is counter-intuitive, unless it is way too small a network, which would suggest that the improved results from an added CNM + fitting are due to extra parameters.**
>
> > **Response to Q4:  Thank you for your valuable comments.**
> * > **why are the CAE images so fuzzy?**
> The input images were constrained to this region of interest and then down-sampled to 31×31 pixels to form the input stimuli set. Thus, these images may not that meaningful and cannot be easy to understand by human. Please see more information about the pre-processing of the image in response to the reviewer ren7th'Q6.
>
>
> * > **Doesn't it give good image reconstruction?**
> CAE has a small network with 3 convolutional layers in the encoder and decoder to reconstruct the image. Please see the setting of **network architectures in Sec 4.1**.
>
>
> * > **This is counter-intuitive, unless it is way too small a network, which would suggest that the improved results from an added CNM + fitting are due to extra parameters.**
> The extra parameter only includes a few weight parameters in the linear mapping function, which influences the neural prediction more than the image reconstruction.  These gains in image reconstruction are due to the inner layer – v1 fitting. Please see the response to **W1**
>
> **Q5: Fig 3: Are you able to add error bars to see if these differences are big? Also, is a higher number of significant neurons better? I believe that in Vi, given a constrained set of images, several neurons might be relatively uninvolved, so higher might be inaccurate. (I don't know).
>
> > **Response to Q5:  Thank you for your comments.**
> * > **Are you able to add error bars to see if these differences are big?**
> There are no error bars because the proportion of insignificant and significant neuron is determined.
> * > **Is a higher number of significant neurons better?**
> Yes, a higher number of significant neurons is better. We compute all the test images on the CNM and DAE-NR. Given a constrained set of images, the CNN part of CNM and DAE-NR can extract the more available feature about the original image and reduce the influence of neural noise. At least, the CNN feature of image in the DAE-NR can better reconstruct the original image than the CNM.
>
> **Q6: Typos: there are various repeated words (eg "however however") , and also a repeated comma.**
>
>
> > **Response to Q6:  Thank you for your valuable comments.** We remove it in our manuscripts.
> >
> *"**However,** DAE and its variants in image reconstruction suffer from the same problem; the parameters have a high degree of freedom."*
>
> **Q7: Page 4: "turns to be" -> "turns out to be"**
>
> > **Response to Q7:**  Thank you for your valuable comments. We rewrite it in our manuscripts.
>
> *"we split the encoder and decoder at the $i$-th layer of DAE. In this way, the architecture of our DAE **turns out to be**"*
>
>
> **Q8: Bottom of page 4: add sub-heading after description of CNM to clarify that the topic is now the hybrid DAE-NR.**
>
> > **Response to Q8:  Thank you for your valuable comments.** We add a sub-heading “DAE-NR” in our manuscripts.
>
> ***DAE-NR:** The loss function of DAE-NR*
>
> **Q9: eqn 3: beta = 1- alpha?**
> > **Response to Q9:** Thank you for your valuable comments. Both beta and alpha are hyperparameters. And beta + alpha is not equal to 1. Please see the setting of the hyperparameters in the **Appendix Table 8-13**.

---

> ### Author Response · Authors · 2022-11-18
> **Response to Reviewer bubd: part 3**
>
> **Q10: Abbreviations could be made clearer, noted at time of first introduction. An example is CNM, which I thought was a typo for most of the paper. Also, what is VQ?**
>
> > **Response to Q10**:  Thank you for your valuable comments. We check the abbreviations in our manuscript and note them at the time of the first introduction. VQ is short name of Vector Quantised [9].
>
> [9] Aäron van den Oord, Oriol Vinyals, and K. Kavukcuoglu. Neural discrete representation learning. In Advances in Neural Information Processing Systems (NeurIPS), 2017.
>
> **Q11: Please note when figures and tables (eg fig 5) are in the Appendix.**
>
> > **Response to Q11:**  Thank you for your valuable comments. We check the figures and tables (e.g. fig 5) are in the Appendix in our manuscript and note they are in the Appendix.
>
> **Q12: "Network architectures": Maybe move the architecture description to the appendix where it can be more complete. The 8C88-9CD77 notation is confusing and leaves out detail.**
>
> > **Response to Q12:**  Thank you for your valuable comments. We move the detailed architecture description to the appendix table 14.
>
> **Q13: Also: what does the "32 x 256" dimension for VQ-VAE relate to?**
>
> > **Response to Q13:**  Thank you for your valuable comments. The dimension of latent embedding space in VQ-VAE is 32 × 256 where 32 is the size of the discrete latent space (i.e., a 32-way categorical), and 256 is the dimensionality of each latent embedding vector e_i [9].
>
> **Q14: page 5" by applying a sparse mask". the explanation of why this is done comes much later. Could you move that explanation so that it follows directly (or precedes)?**
> > **Response to Q14: ** Thank you for your valuable comments. we move it to the front.
>
> *Applying a sparse mask, the readout layer pools the output of the convolutional layer (i.e., hi) on each neuron.*
>
> **Q15: Table 2 and 5: How do you restrict the CAE image reconstruction to use only a single layer? This is not described in 3.2.**
>
> > **Response to Q15:**  Thank you for your comments.
> > In our study, the representational similarity of V1 neurons is brought to a specific layer of the CAE encoder (hi, i ∈ [1, 2, 3, 4]) with **the sparsity constraint for artificial neurons** in this layer (see in Sec 3.2)
>
> *"Previous studies have shown that the responses of V1 neurons to natural stimuli are sparse, and the activity of neural populations with higher sparseness exhibits greater discrimination against natural stimuli. (Vinje & Gallant, 2000; Weliky et al., 2003; Froudarakis et al., 2014; Yoshida & Ohki, 2020). Likewise, (Zhuang et al., 2017) has reported that the resemblance between the representation of biological neurons and artificial neurons in higher convolutional layers exists only under the sparsity constraint on the CNN, regardless of any other factors (e.g., , model structure, training algorithm, receptive field size, and property of training stimuli). In our study, the representational similarity of V1 neurons is brought to a specific layer of the CAE encoder (hi, i ∈ [1, 2, 3, 4]) with the sparsity constraint for artificial neurons in this layer."*
>
> **Q16:Table 5: do you have error bars (std devs)? Related: what is the appropriate number of decimal places?**
> > **Response to Q16:**  Thank you for your comments. In our study, there are no error bars because the proportion of insignificant and significant neuron is determined. Please see the response to **W1**

---

> > ### Comment · Reviewer_bubd · 2022-11-23
> > **response to authors' comments**
> >
> > My true thanks to the authors for responding carefully to my (and other reviewers') comments.
> >
> > I find myself out of depth tracking the nuances of the various reviewers' comments and authors' responses.
> > So while I am still not convinced by the overall gist of the paper, I recognize that this may be my problem, not the paper's :)
> >
> > I wish to defer to the other reviewers as to disposition. For example, vnH9 and n7th appear to bring more technical experience and granular insight to bear on this particular paper than I can.

---

### Official Review · Reviewer_vnH9 · 2022-10-25

**Confidence:** 4
**Correctness:** 3
**Technical Novelty And Significance:** 2
**Empirical Novelty And Significance:** 3
**Recommendation:** 5

**Clarity, Quality, Novelty And Reproducibility:**

Clarity: The paper is readable but can be improved signficantly with a rewrite. I would encourage authors to reduce the number of acronyms used in the paper to help readability

Quality: The quality of experiments is good on the DNN side. The experiment however is done only on one dataset of brain recordings and how it compares to state of the art models in neural activity prediction on Brainscore.

Originality: The idea to use neural data as an auxillary loss along with an image related task is not new. However, here it is applied to a new problem and shows improvement in both reconstruction and representational similarity with biological neurons. The results therefore are new.



**Strength And Weaknesses:**

Strengths:
1. The idea although simple improves the performance on both the tasks.
2. Experiments seem sufficient enough for validating the proposed idea
3. The idea is evaluated using different autoencoders suggesting generalizability of the claims.

Weakness:
1. Minor: the idea presented here is simple multitask network. I am not sure if this is a "framework"
2. In Table 2: if CAE is trained without any brain data why is there any variation in result of region 1, 2 and 3?
3. What is the difference between NR and FR.? there are lots of acronyms in the paper making it difficult to follow. I would suggest unifying the acronyms as in Tables only CAE is used but in the text there are multiple mentions of DAE-NR. It was only possible to figure out after multiple reads.
4. There are lots of publicly available datasets from Brainscore (Schrimpf et al. 2018) and others. Why was this particular dataset chosen?
5. Generalizability is only shown in V1, using other public datasets maybe it might have made sense to show that approach is generalizable to different brain regions as well.

**Summary Of The Paper:**

In this paper, an image autoencoder is trained with two loss functions: 1. Image reconstruction loss and 2. Poisson loss to optimize representation similarity between artificial and biological neurons.

The authors show that by training in the above way:
1. Image reconstruction is better than the case without Poisson loss
2. The model shows high resemblance between artificial/biological neurons as compared to standard end to end models

**Summary Of The Review:**

The idea although not new is applied in a new way to improve both the performance on image reconstruction task and neural response similarity task.

Due to limited testing on brain dataset even though there are lots of publicly available benchmarks I am inclined towards giving it a borderline reject.

However, if authors provide a valid reason to not use those benchmarks or provide new results I will be happy to update my rating.

---

> ### Author Response · Authors · 2022-11-18
> **Response to Reviewer vnH9**
>
> Thanks for your comments that **this work improves neural similarity and image reconstruction performance**. We have made revisions to our paper based on your feedback and addressed your concerns in more detail in the comment below.
>
> **Q1: Minor: the idea presented here is simple multitask network. I am not sure if this is a "framework"**
>
> > **Response to Q1:**  Thank you for your comments. We **agree with the reviewer** that our idea is in line with the multitask network. We call it ‘framework’ for the way to add biological information into artificial neural networks is general, and not limited to certain type of network architecture. We therefore consider it a simple yet effective framework for unifying representation similarity and function in biological and artificial neural networks. We implement different network architectures (i.e., CAE, VAE, VQ-VAE) in our framework. Please see Table 1.
>
> **Q2: In Table 2: if CAE is trained without any brain data why is there any variation in result of region 1, 2 and 3?**
> > **Response to Q2:**  Thank you for your comments. Indeed, CAE is trained without any brain data. The **variation may come from the different**  training and testing stimuli set for regions 1, 2 and 3.  So the result of regions 1, 2 and 3 are the difference.
>
> **Q3: What is the difference between NR and FR? there are lots of acronyms in the paper making it difficult to follow. I would suggest unifying the acronyms as in *Tables only CAE is used but in the text there are multiple mentions of DAE-NR*. It was only possible to figure out after multiple reads.**
>
> > **Response to Q3:**  Thank you for your comments.
> * > **What is the difference between NR and FR?**
> We are sorry about the **acronyms**. NR is short for neural response, which is the response of the biological neurons to the stimuli. **DAE-NR refers to the DAE with information of neural response** (our model).
> * > **There are lots of acronyms in the paper making it difficult to follow.**
> We are so sorry for the acronyms in the paper making it difficult to follow. We unify the acronyms as in Table 1.
>
> **Q4: There are lots of publicly available datasets from Brainscore (Schrimpf et al. 2018) and others. Why was this particular dataset chosen?**
> > **Response to Q4:**  Thank you for your suggestion. The following are the reason why we do not choose to use Brainscore.
> >   a). Previous work on Brainscore used a large model pre-trained on ImageNet (e.g., AlexNet) as the activation model. When converting the large model into a brain model, all the weights of variables in the activation model will be frozen. Only the variables in the Readout Layer need to be trained. Thus, the number of training samples in Brainscore is sufficient for their training. However, in this work, we cannot train our network on a large dataset due to the training procedure of our model requires images and the related neural data at the same time.
> >   b). The size of images in Brainscore (256x256x1) is much larger than the size of our chosen downsampled dataset (32x32x1). Training on the Brainscore with our network architectures will cause significant underfitting.
> >   c). We have tried to downsample the Brainscore image data, but this will result in a mosaic-like appearance of the downsampled image without any shape or texture features.
>
>
> **Q5: Generalizability is only shown in V1, using other public datasets maybe it might have made sense to show that approach is generalizable to different brain regions as well.**
>
> > **Response to Q5:** Thank you for your suggestion. The features captured by V1 are relatively low-level image features, while V2, V4, and IT capture more abstract features. Thus, a larger amount of data is needed to support us to prove whether V2, V4, and IT have the generalization ability. However, it is a pity that the amount of existing data is hard to support us to do this experiment.

---

### Decision · Program_Chairs · 2023-01-20

**Decision:**

Reject

**Justification For Why Not Higher Score:**

The significance of the work is limited.

**Justification For Why Not Lower Score:**

NA

**Metareview: Summary, Strengths And Weaknesses:**

This paper shows that optimizing a CAE model to fit neural data and reconstruct input images at the same time improves the performance on both tasks. There is general consensus that the finding that integrating neural response into deep autoencoder models can improve its performance is interesting. But there is also general consensus (with the exception of one review)  that the significance of the work remains limited as previous work has already shown the benefits of jointly training/regularizing vision tasks with neural data. Some of it is cited (and additional work that could be cited include: https://arxiv.org/abs/2211.07885).

The contribution of this work is really only to show how neural data help an auto-encoder which appears limited in terms of significance.